# 326K at E Protein Is Critical for Mammalian Adaption of TMUV

**DOI:** 10.3390/v15122376

**Published:** 2023-12-01

**Authors:** Xingpo Liu, Dawei Yan, Shan Peng, Yuee Zhang, Bangfeng Xu, Luzhao Li, Xiaona Shi, Tianxin Ma, Xuesong Li, Qiaoyang Teng, Chunxiu Yuan, Qinfang Liu, Zejun Li

**Affiliations:** 1Shanghai Veterinary Research Institute, Chinese Academy of Agricultural Sciences, Shanghai 200241, China; xingpo.liu@cansinotech.com (X.L.); yandawei@shvri.ac.cn (D.Y.); zye@shinhwa.cn (Y.Z.); xubangfeng@shvri.ac.cn (B.X.); luzhao.li@wur.nl (L.L.); liuyx@ustc.edu.cn (X.S.); 2023390141@gzhmu.edu.cn (T.M.); lixuesong@hainanu.edu.cn (X.L.); tengqy@shvri.ac.cn (Q.T.); yuanchx@shvri.ac.cn (C.Y.); 2School of Chemistry and Chemical Engineering, University of Jinan, Jinan 250022, China; shan.peng@3dbiooptima.com

**Keywords:** Tembusu virus, envelope protein, mutation, inflammatory, RIG-I-IRF7

## Abstract

Outbreaks of Tembusu virus (TMUV) infection have caused huge economic losses to the poultry industry in China since 2010. However, the potential threat of TMUV to mammals has not been well studied. In this study, a TMUV HB strain isolated from diseased ducks showed high virulence in BALB/c mice inoculated intranasally compared with the reference duck TMUV strain. Further studies revealed that the olfactory epithelium is one pathway for the TMUV HB strain to invade the central nervous system of mice. Genetic analysis revealed that the TMUV HB virus contains two unique residues in E and NS3 proteins (326K and 519T) compared with duck TMUV reference strains. K326E substitution weakens the neuroinvasiveness and neurovirulence of TMUV HB in mice. Remarkably, the TMUV HB strain induced significantly higher levels of IL-1β, IL-6, IL-8, and interferon (IFN)-α/β than mutant virus with K326E substitution in the brain tissue of the infected mice, which suggested that TMUV HB caused more severe inflammation in the mouse brains. Moreover, application of IFN-β to infected mouse brain exacerbated the disease, indicating that overstimulated IFN response in the brain is harmful to mice upon TMUV infection. Further studies showed that TMUV HB upregulated RIG-I and IRF7 more significantly than mutant virus containing the K326E mutation in mouse brain, which suggested that HB stimulated the IFN response through the RIG-I-IRF7 pathway. Our findings provide insights into the pathogenesis and potential risk of TMUV to mammals.

## 1. Introduction

Tembusu virus (TMUV) belongs to the Flavivirus genus of the Flaviviridae family, which includes over 70 viruses such as Japanese encephalitis virus (JEV), Zika virus (ZIKV), West Nile virus (WNV), forest encephalitis virus (FEV), and tick-borne encephalitis virus (TBEV) [1,2]. Flavivirus has a single-stranded positive-sense RNA genome that encodes three structural proteins (capsid, precursor membrane protein, and envelope protein) and seven non-structural proteins (NS1, NS2A, NS2B, NS3, NS4A, NS4B, and NS5) [2,3]. The envelope protein (E) of flavivirus, the major surface protein of the viral particle, plays a significant role in virulence and host specificity. It participates in key steps of the viral life cycle, including receptor attachment, entry, membrane fusion, and assembly [4,5,6,7].

TMUV was first isolated from *Culex* spp. in Malaysia in 1955 [8]. In 2010, TMUV caused the first epidemic in ducks in China and then rapidly spread to several Asian countries, resulting in huge economic losses to the poultry industry [9,10,11]. Moreover, TMUV was found to infect other species, such as chickens, sparrows, and geese [12,13], and replicate efficiently in both human nerve and liver cell lines [14]. A surveillance study showed that 71.9% of the investigated duck farm workers were seropositive to TMUV, and TMUV RNA was detected in 47.7% of oral swabs [11,15]. Although the TMUV-positive people did not show any clinical signs, it is still unclear whether TMUV poses a potential threat to public health. Unlike other flaviviruses, which are mainly transmitted by mosquitoes or ticks among hosts, our previous studies have shown that TMUV is airborne-transmissible in ducks [16]. Given the airborne transmissibility and the severe disease outcome of TMUV in birds, it is necessary to further explore the potential threat and pathogenesis of TMUV in mammals. The E protein of flavivirus, the major surface protein of the viral particle, plays a predominant role in virulence and host-specific adaptation since it is critical for receptor attachment, entry, membrane fusion, and viral assembly [17]. Previous studies have shown that a single mutation in the E protein altered the virulence and tissue tropism of flavivirus in vivo. The G306E mutation in the E protein reduced the affinity between the virus and the receptor, resulting in loss of neurotoxicity of JEV [18]. The 304R and 367T residues in the E protein were proved to be important for the pathogenicity of TMUV in ducks [17,19]. All the studies indicated that the E gene is critical for the pathogenesis of flaviviruses.

Similar to JEV, WNV, and Usutu virus, nonpurulent encephalitis caused by TMUV was the main cause of death in ducks [20]. Previous studies have shown that TMUV does not infect mice through intranasal (i.n.) or subcutaneous (s.c.) inoculation routes [21,22]. Recently, we found that a duck TMUV HB strain infected mouse CNS via i.n. inoculation route and caused encephalitis and deaths in five-week-old female BALB/c mice. The purpose of this study is to reveal the molecular mechanism of pathogenicity of TMUV HB in mice through i.n. inoculation routes. We explored the path of the virus entering the brain through nasal inoculation, and identified the key amino acid residues that determine the virus virulence through reverse genetic technology. Furthermore, we identified the signaling pathway leading to mouse encephalitis by detecting the expression levels of inflammatory cytokines and type I interferon in the brains of infected mice.

## 2. Materials and Methods

### 2.1. Ethics Statement

All animal experiments were performed in accordance with the Chinese Regulations of Laboratory Animals, the Guidelines for the Care of Laboratory Animals (Ministry of Science and Technology of the People’s Republic of China), and Laboratory Animal Requirements of Environment and Housing Facilities (GB 14925-2010, National Laboratory Animal Standardization Technical Committee). The protocol (SHVRI-SZ-20191227-01) used in the study was approved by the Animal Care Committee of the Shanghai Veterinary Research Institute.

### 2.2. Viral Infection

Baby Syrian hamster kidney (BHK-21) cells (ATCC) were cultured in Dulbecco’s modified Eagle medium (DMEM) (BI, USA) containing 5% fetal bovine serum (FBS, PAN Biotech, Aidenbach, Germany), 100 U/mL penicillin, and 100 μg/mL streptomycin (Invitrogen, Waltham, MA, USA) at 37 °C in a 5% CO_2_ humidified incubator.

TMUV HB strain and FX2010 (GenBank: MH414568.1, FX2010 is a TMUV strain isolated from diseased ducks in China in 2010) were isolated from ducks and purified three times in specific-pathogen-free chicken embryonated eggs by a limiting dilution method. All of the rescued viruses were propagated once on BHK-21 cells, aliquoted, and stored at −80 °C.

### 2.3. Sequence Analysis

The viral RNAs were extracted using TIANamp Virus RNA Kit (Tiangen, Beijing, China), and cDNAs were synthesized with specific reverse transcription primers (Appendix A). Nine pairs of primers targeting the highly conserved sequences of TMUV were used to amplify nine overlapping fragments covering the whole genome of the HB strains, respectively. The sequencing results were analyzed using DNASTAR software. The reference virus strain information has been Appendix A.

### 2.4. Generation of Mutant Viruses

Four plasmids, pHBT7-1-976, pHB942-2459, pHB2433-3831, and pHB3656-10991, covering the whole genome of HB were generated as described previously [23]. Plasmids containing single-site mutation on the E or NS3 protein were generated through two-step PCR methods based on the background of the plasmids pHB942-2459 and pHB3656-10991, which contained the E and NS3 gene of HB.

The full-length cDNAs with T7 promoter were generated by two rounds of PCR using High Fidelity DNA polymerase pfu (Invitrogen, Waltham, MA, USA) as reported previously [24]. Four overlapping fragments were first amplified by PCR using the four plasmids (pHBT7-1-956, pHB942-2459, pHB2433-3831, and pHB3656-10991) as templates. To rescue mutant viruses, pHB942-2459 or pHB3656-10991 was replaced with the corresponding modified plasmid. The full-length cDNAs were transcribed into infectious viral RNAs using an mMESSAGEmMACHINE^®^ T7 kit (Invitrogen, Waltham, MA, USA). The transcripts were purified by lithium chloride and transfected into BHK-21 cells with Lipofectamine LTX and Plus Reagent (Invitrogen, Waltham, MA, USA) at a dose of 5 µg per well on a 6-well plate. The cell culture medium was changed to the DMEM containing 2% FBS at 8 h post transfection. When cytopathic effects appeared in the transfected cells, the supernatants were collected, and rescued viruses were amplified on DF-1 cells, aliquoted, and stored at −80 °C. The rescued viruses were identified by genome sequencing as described above.

### 2.5. Virus Titration

To determine viral loads in mice tissues, the samples were weighed and homogenized in sterile PBS to yield 1:1 (mL/g) homogenates. Tissue homogenates were clarified by centrifugation at 12,000× *g*, 4 °C for 10 min; 10-fold serially diluted supernatants were titrated onto 80% confluent BHK-21 cells in 96-well plates at 37 °C for 2 h. After adsorption, cells were washed and cultured with DMEM with 2% FBS in a humidified chamber with 5% CO_2_ at 37 °C. The lower limit of virus detection was 0.5 log10 TCID_50_ per 0.1 g tissue. The virus titer was calculated by the method of Reed and Muench.

### 2.6. TaqMan Probe-Based Quantitative Real-Time RT-PCR (qRT-PCR)

Total RNAs were extracted from tissues of HB-infected mice using TRIzol reagent (Thermo Fisher Scientific, Waltham, MA, USA) following the manufacturer’s instructions. Reverse transcription was performed using a PrimeScript RT reagent kit with gDNA Eraser (TaKaRa, Kusatsu, Japan) according to the manufacturer’s instructions. The copies of viral E gene were determined using TaqMan probe-based qRT-PCR, as described previously [25]. To test the level of cytokine expression, total RNA was extracted from brain tissue samples with TRIzol reagent (Thermo Fisher Scientific, Waltham, MA, USA) according to the manufacturer’s instructions. RNA was further reverse transcribed into cDNA using a PrimeScript RT reagent kit with gDNA Eraser (TaKaRa, Kusatsu, Japan). The mRNA levels of IL-1β, IL-2, IL-6, IL-8, TNF-α, IFNs, RIG-I, MDA5, TLR3, TLR7, IRF3, and IRF7 were determined by qRT-PCR with SYBR Premix Ex Taq (TaKaRa, Kusatsu, Japan) and fold changes in gene expression were calculated with the 2^−ΔΔCT^ method. Information on primers used is shown in Appendix A.

### 2.7. Measurement of Proinflammatory Cytokines and IFN-α/β Level

Protein levels of proinflammatory cytokines and IFN-α/β in brain homogenate supernatants were measured with enzyme-linked immunosorbent assay (ELISA) kits according to the manufacturer’s instructions. The ELISA kits for detection of mouse IFN-α and IFN-β were purchased from Cusabio (CUSABIO, Wuhan, China). The ELISA kits for detection of mouse IL-1β, IL-2, IL-6, TNF-α and IL-8 were purchased from Biolegend (Biolegend, San Diego, CA, USA).

### 2.8. Mouse Experiments

#### 2.8.1. Neuroinvasiveness Assays

To test the neuroinvasiveness of TMUV HB and FX2010 in mice, five-week-old BALB/c mice were randomly divided into three groups and inoculated intranasally with 10^5.0^ TCID_50_ of HB or FX2010. Mock-infected mice received PBS. Mice were monitored daily for the appearance of symptoms for 14 days post infection. To further determine the virus distribution in mice, brain, lung, and nasal turbinate were collected and tested at 3, 5, and 7 dpi. The brains were collected, fixed with 4% paraformaldehyde for 48 h, and processed according to standard immunohistochemistry methods. Viral antigens in the brain were detected by immunohistochemistry (IHC); a monoclonal antibody (MAb) 1F5 against E protein was used in the IHC experiment.

#### 2.8.2. Olfactory Epithelium Assays

Given that the previously isolated TMUV strain (reference strain FX2010) did not show any clinical symptoms in mice by i.n. inoculation routes, the TMUV HB strain isolated in this study was able to produce high viral titers in the mouse brain through i.n. inoculation routes, leading to mouse death. In this study, two TMUV strains with obvious pathogenic differences in mice were selected to explore the molecular mechanism of TMUV HB causing mouse death by i.n. inoculation routes. Olfactory epithelium connects the nasal cavity and the brain, making it one of the main ways for flavivirus to infect the brain, according to previous studies [26]. To explore whether HB utilizes the olfactory epithelium to enter the brain, five-week-old female BALB/c mice were randomly divided into four groups (HB, ZnSO_4_-HB, ZnSO_4_, MOCK), in which the infection of ZnSO_4_-HB and ZnSO_4_ groups was accomplished by bilateral ZnSO4 intranasal perfusion perfused with 10% ZnSO_4_ at a volume of 30 µL to destroy the olfactory neuron, respectively. The olfactory function was evaluated by a sensitive olfactory-mediated behavioral test, that is, the ability of mice to discriminate between water and butanol within a T-maze test. The apparatus was a T-shaped maze with two dark arms each containing a watch-glass with a filter paper soaked with either distilled water (nonodorant) or a butanol solution (known to be a repulsive odorant for mice) [27]. Six of the mice were placed in a start-box at the end of the stem in turn, allowed to move freely for 5 min, and then tested randomly. Each mouse was tested three times with 30 min intervals and the positions of water and butanol were placed alternately. Discrimination ability was evaluated as the percentage of entrances in the water-containing arm. The HB and ZnSO_4_-HB groups of mice were challenged with HB on day 4 after ZnSO4 i.n. perfusion. Due to the fact that mice are more sensitive to the odor of 15% high-concentration butanol than 10% low-concentration butanol, we used 15% butanol in olfactory nerve damage experiments to examine olfactory-mediated behavior. Mice survival (n = 5 per group) and weight changes (n = 3 per group) were recorded daily and viral titers in the brains of the mice were determined by qRT-PCR at 7 dpi.

#### 2.8.3. Determination of Key Amino Acid Responsible for the Virulence of TMUV HB

To identify the key amino acids that affect the neuroinvasiveness of the TMUV HB strain, five-week-old female BALB/c mice were inoculated with 10^5.0^ TCID_50_ (the inoculation dose has been optimized) of parental or mutant viruses via i.n. inoculation routes. Groups of three mice were observed daily for clinical signs and the weight changes were recorded. Three mice were euthanized at 4, 6, and 8 dpi. The brains were collected for virus detection. The brains were collected and homogenized in PBS, and the viral titer in the supernatant was determined on BHK-21 cells.

To determine the contribution of 326K and 519T to the neurovirulence of TMUV in mice, five-week-old BALB/c mice were infected through the intracranial (i.c.) route with the wtHB-E_326K_NS3_519T_, smHB-E_326E_NS3_519T_, smHB-E_326K_NS3_519A_, and dmHB-E_326E_NS3_519A_ viruses containing 10^1.0^, 10^2.0^, 10^3.0^ or 10^4.0^ TCID_50_. In each group, three inoculated mice were euthanized at 4, 6, and 8 dpi by CO_2_ inhalation, tissue samples from brains were collected for viral titration, and three mice were observed daily for clinical signs and weight loss.

To further explore the mechanism of 326K to the virulence of TMUV, five-week-old BALB/c mice were inoculated through i.c. route with the wtHB-E_326K_NS3_519T_, smHB-E_326E_NS3_519T_, smHB-E_326K_NS3_519A_, and dmHB-E_326E_NS3_519A_ viruses containing 10^3.0^ TCID_50_. The brains of three mice in each group were collected at 4, 6, and 8 dpi for the detection of viral loads, proinflammatory cytokines, and IFNs.

To verify whether type-I IFN affected the pathogenesis of TMUVs in mice, five-week-old BALB/c mice were inoculated i.c. with 10^3.0^ TCID_50_ of wtHB-E_326K_NS3_519T_ or smHB-E_326E_NS3_519T_, respectively. Then, all six infected mice were inoculated i.c. with commercial human IFN-β (3000 U) at 0, 3, and 5 dpi, and death of the mice was recorded. Human IFN-β was purchased from Peprotech (Peprotech, Cranbury, NJ, USA).

### 2.9. Statistical Analysis

All statistical analyses were performed with GraphPad Prism version 7.0 (GraphPad Software, Inc., San Diego, CA, USA). Data are expressed as means with standard deviations (SD). For viral titers and inflammatory cytokines, an unpaired Student’s *t*-test was used to determine statistically significant differences. Statistically significant differences in body weight were determined using two-way ANOVA multiple comparisons. A *p* value < 0.05 was considered significant.

## 3. Results

### 3.1. A TMUV HB Strain Is Neuroinvasive to Mice through Intranasal Infection

According to previous studies, TMUV causes systemic infection and encephalitis in ducks, but not in mammalian models [17]. In this study, a TMUV HB isolate from diseased ducks caused 80% mortality in five-week-old female BALB/c mice inoculated through i.n. route. All six infected mice showed clinical signs from 6 dpi, such as hunching and ruffled fur, whereas mice infected with the duck TMUV representative strain FX2010 did not exhibit any clinical symptoms. The mice infected with the HB strain experienced significant weight loss (*p* < 0.01) compared with the mock and FX2010 groups (Figure 1A,B). To determine the tissue tropism of the virus in mice, we collected the brain, lungs, and turbinates at different time points for virus load testing. The results showed that HB was detected in the brain, lung, and nasal turbinate, with the viral load in brain being significantly higher than in other organs. In contrast, the FX2010 could only be detected in the lung and nasal turbinate, but not in the brain. (Figure 1C). Immunohistochemistry results showed that the TMUV antigens were detected in the brain of mice infected by HB, but not the FX2010 (Figure 1D). All the results indicate that the TMUV HB strain is neuroinvasive in mice.

### 3.2. Olfactory Neurons Are the Bridge for TMUV HB to Invade the Mouse CNS

Olfactory epithelium is a specialized tissue that contains olfactory neurons (ON) in the nasal cavity. It connects the nasal cavity and the brain, making it one of the main ways for flavivirus to infect the brain, according to previous studies [26,28]. To investigate whether the HB strain uses the olfactory pathway to enter the mouse brain during i.n. infection, olfactory neurons of BALB/c mice were destroyed. Mice were intranasally perfused with ZnSO_4_ to interrupt the connection between nasal cavity and CNS as described previously [29]; then, the treated mice were infected with HB through i.n. To verify the olfactory neurons’ destruction in vivo, we evaluated the ability of mice to distinguish between water and butanol within a T-maze test as described previously [23] (Figure 2A). Untreated mice showed a preference for entering the water-containing arm (day 0: 88%), while the ability of the treated mice to distinguish between water and butanol decreased after ZnSO_4_ treatment. Mice showed the most damage to olfactory function on day 4 after treatment (approximately 55% entrances in water-containing arm). The mice that received ZnSO_4_ pretreatment (HB-ZnSO_4_) showed reduced mortality from 80% to 40% after infection with HB (Figure 2B). The group infected with HB alone showed more weight loss than the HB-ZnSO_4_ group. Both groups of infected mice showed symptoms such as hunching and congested fur, but those of the HB-ZnSO_4_ group were significantly milder (Figure 2C). Furthermore, the destruction of olfactory epithelium leads to a reduction of approximately 128 times of the virus load in mouse brain (*p* < 0.05). (Figure 2D). These results indicate that the TMUV HB strain enters the brain through olfactory epithelium.

### 3.3. Both E 326K and NS3 519T Residues Contribute to the Neuroinvasiveness of TMUV HB

To investigate the mechanism underlying the neuroinvasiveness of HB in mice, the whole genome of HB was sequenced and aligned with all available genome sequences of TMUVs deposited in GenBank. The alignment result showed that the HB strain (belonging to cluster 2.2^42^) had two unique amino acid residues, 326K in the E protein and 519T in the NS3 protein, while most TMUV strains had 326E and 519A. Interestingly, amino acids 326K and 519T were only found in TMUV from cluster 2.2, but not in other clusters (Appendix A). To determine the impact of these two residues on the neuroinvasive ability of HB, one double-site mutant virus (dmHB-E_326E_NS3_519A_) with E K326E and NS3 T519A substitutions, and two single-site mutant viruses (smHB-E_326E_NS3_519T_ and smHB-E_326K_NS3_519A_) were rescued in the background of the wild type HB (wtHB-E_326K_NS3_519T_).

The neuroinvasiveness of the four viruses was evaluated in mice. The results showed that BALB/c mice infected by i.n. with the wtHB-E_326K_NS3_519T_ and smHB-E_326K_NS3_519A_ had mortality rate of 80% and 20%, respectively, while no death was observed in the mice infected with the smHB-E_326E_NS3_519T_ and dmHB-E_326E_NS3_519A_. Compared with wtHB-E_326K_NS3_519T_ and smHB-E_326E_NS3_519T_, the K to E mutation at 326 of E protein led to a reduction in the mortality of infected mice from 80% to 0. In addition, smHB-E_326K_NS3_519A_ and dmHB-E_326E_NS3_519A_ showed that the mutation of K to E at position 326 of E protein also reduced the mortality of infected mice from 20% to 0. This result indicates that E protein 326K plays a dominant role in the neuroinvasiveness of TMUV HB in mice (Figure 3A). Furthermore, mice infected with the wtHB-E_326K_NS3_519T_ exhibited a significant decrease in weight (*p* < 0.05) compared to other groups, including the smHB-E_326K_NS3_519A_ (Figure 3B). Viral titration results showed that the virus was detected in the brains of all three mice infected with wtHB-E_326K_NS3_519T_ at 4, 6, and 8 dpi, and in one mouse in the smHB-E_326K_NS3_519A_-infected group at 6 and 8 dpi, while the virus was not detected in smHB-E_326E_NS3_519T_- and dmHB-E_326E_NS3_519A_-infected groups (Figure 3C). The results demonstrate that K326E mutation in E protein abolishes the neuroinvasiveness of TMUV HB in mice; T519A mutation attenuates the smHB-E_326K_NS3_519A_ compared with wtHB-E_326K_NS3_519T_. All the results suggested that both the E 326K and NS3 519T residues contribute to the neuroinvasiveness of TMUV HB, and E 326K is more critical.

### 3.4. The 326K Residue of E Protein Plays a Major Role in the Neurovirulence of TMUV

To further investigate the contribution of 326K and 519T to the neurovirulence of TMUV, we examined the median lethal dose (LD_50_) of mice infected with four viruses through i.c. The results showed that wtHB-E_326K_NS3_519T_ caused 100% mortality in mice inoculated with 10^2.0^ or 10^3.0^ TCID_50_ viruses, with an LD_50_ of 10^1.5^ TCID_50_, while no mortality occurred in smHB-E_326E_NS3_519T_-infected mice at a dose of 10^4.0^ TCID_50_ (Figure 4A,C). The LD_50_ of smHB-E_326K_NS3_519A_ in mice was 10^1.5^ TCID_50_, which is the same as that of wtHB-E_326K_NS3_519T_ (Figure 4A,B). In contrast, the LD_50_ of dmHB-E_326E_NS3_519A_ was lower than that of smHB-E_326E_NS3_519T_, which did not cause mouse death at a dose of 10^4.0^ TCID_50_ (Figure 4C,D). Furthermore, the LD_50_ of dmHB-E_326E_NS3_519A_ was also 5–6 times higher than that of smHB-E_326K_NS3_519A_ (Figure 4B,D). The body weight change showed that the weight of all infected mice decreased at 4 dpi, whereas the body weight of mice infected with the smHB-E_326E_NS3_519T_ gradually recovered from 7 dpi (Appendix A). All the results indicated that the 326K residue of E protein plays a more critical role in the neurovirulence of TMUV HB strain than the 519T residue in NS3.

### 3.5. The TMUV HB Strain Induced Higher Levels of Inflammatory Response Than the Mutant HB with K326E Mutation in Mouse Brain

To further explore the mechanism underlying the neurovirulence of TMUV HB, BALB/c mice were infected through i.c. with the wtHB-E_326K_NS3_519T_, smHB-E_326E_NS3_519T_, smHB-E_326K_NS3_519A_, and dmHB-E_326E_NS3_519A_ viruses. The wtHB-E_326K_NS3_519T_ replicated to higher titers than the smHB-E_326E_NS3_519T_ at 6 and 8 dpi, while smHB-E_326K_NS3_519A_ and dmHB-E_326E_NS3_519A_ has similar replication levels in the mouse brain (Figure 5A). Groups of infected mice showed hunching, congested fur, paralysis, and neurological symptoms such as tremor. We evaluated the mRNA and protein levels of proinflammatory cytokines in brains of the mice infected with the wtHB-E_326K_NS3_519T_, smHB-E_326E_NS3_519T_, smHB-E_326K_NS3_519A_, and dmHB-E_326E_NS3_519A_. The levels of IL-1β, IL-6, and IL-8 in the brains of wtHB-E_326K_NS3_519T_-infected mice were significantly higher than those of the smHB-E_326E_NS3_519T_ group at 6 dpi and 8 dpi (Figure 5; Appendix A). Further, we checked the pathological changes in the brain tissue of the infected mice. Inflammatory cell infiltration and vascular cuff around the cerebral vessels were observed in the brain pathological sections of wtHB-E_326K_NS3_519T_-infected mice, but rarely in smHB-E_326E_NS3_519T_-infected mice (Appendix A). These results suggest that the E to K mutation may trigger an excessive inflammatory response in the mouse brain, thereby enhancing the neurovirulence of the virus.

The analysis of IFN-α/β induction at both mRNAs revealed that the wtHB-E_326K_NS3_519T_ and smHB-E_326K_NS3_519A_ strains induced significantly higher levels of IFN-α/β production than smHB-E_326E_NS3_519T_ and dmHB-E_326E_NS3_519A_ in brain at 6 dpi and 8 dpi (Figure 5). To investigate the role of type-I IFN in the neurovirulence of TMUV HB, human recombinant IFN-β (3000 U) was administered intracranially in mice at various times after infection with the wtHB-E_326K_NS3_519T_ and mutant smHB-E_326E_NS3_519T_. The results showed that the recombinant IFN-β accelerated the death of mice in the wtHB-E_326K_NS3_519T_ group compared with the untreated groups (Appendix A). Similarly, the smHB-E_326E_NS3_519T_ caused 40% mortality in mice treated with the IFN-β, while none of the mice died in the smHB-E_326E_NS3_519T_-alone-infected group (Appendix A). All the results suggest that excessive inflammatory response, coupled with overexpression of type-I IFN, collectively contributed to the virulence of the wtHB-E_326K_NS3_519T_ in mice.

### 3.6. TMUV HB Significantly Upregulated the RIG-I-IRF7 Signaling Pathway in Mouse Brains

To explore the pathway by which TMUV HB stimulated proinflammatory cytokines and IFNs, the transcript levels of RIG-I, MDA5, TLR3, TLR7, IRF3, and IRF7 were examined in the brains of mice infected with the wtHB-E_326K_NS3_519T_, smHB-E_326E_NS3_519T_, smHB-E_326K_NS3_519A_, and dmHB-E_326E_NS3_519A_. The transcriptional level of RIG-I induced by the wtHB-E_326K_NS3_519T_ was significantly higher than that induced by the smHB-E_326E_NS3_519T_ at 4 and 6 dpi. Similarly, MDA5 also showed significant differences at 6 dpi. The transcript level of RIG-I in the smHB-E_326K_NS3_519A_-infected mice was also significantly higher than that of the dmHB-E_326E_NS3_519A_ at 4 dpi, and MDA5 also showed significant differences at 4, 6, and 8 dpi (Figure 6A). Furthermore, the level of IRF7 but not IRF3, TLR3, and TLR7 stimulated by the wtHB-E_326K_NS3_519T_ and smHB-E_326K_NS3_519A_ was significantly upregulated compared with the smHB-E_326E_NS3_519T_ or dmHB-E_326E_NS3_519A_ (Figure 6B,C). All the results suggested that the residue 326K of E protein is critical for the stimulation of RIG-I -IRF7-dependent IFN response in wtHB-E_326K_NS3_519T_-infected mouse brains.

## 4. Discussion

Accumulating evidence suggests that TMUVs have an expanding host range, raising concerns about a potential threat to public health [11,18,30]. Studies have demonstrated the ability of TMUV to infect a panel of cell lines from various mammals, including mice, nonhuman primates, and humans, indicating a broad range of infectivity [15]. In particular, the proliferation profile of TMUV in human neuroblastoma SH-SY5Y cells suggests that it has the potential to infect the nervous system [15]. The TMUV HB strain possesses the ability to infect the CNS of mice via the nasal cavity. The blood–brain barrier (BBB) can be bypassed by viruses via three primary routes: hematogenous, less fortified barrier structures (such as the choroid plexus and circumventricular organs), and peripheral nerves (including the olfactory nerve) [28]. The olfactory nerve is the shortest nerve in the CNS, connecting the nasal olfactory epithelium and the olfactory bulb [31]. In this study, partially destroyed olfactory epithelium attenuated TMUV HB infection in mice and partially blocked the entry of the virus into brain, which suggested that TMUV HB primarily enters the mouse brain via olfactory neurons. Previous studies have confirmed that olfactory neurons provide a rapid and efficient pathway for HSV-1 and other respiratory viruses to enter the CNS [32]. WNV, a member of the Flavivirus family, invades CNS mainly through olfactory neurons [33]. Flavivirus family members are mostly zoonotic; the TMUV HB strain infected the mouse CNS through the olfactory epithelium, suggesting that TMUV has potential to cross the host barrier and infect mammals. This study emphasizes the potential threat of TMUV to public health.

The pathogenicity of the Flavivirus is attributed to genetic modifications in multiple genes and even noncoding regions [17,19,34,35]. The E protein, which is the primary virulence-associated protein of flaviviruses, plays an important role in host-specific adaptation, cell tropism, virus attachment, and release [36,37,38]. For instance, the 304R and 367K residues at the E protein contribute to the attenuation of TMUV and are associated with enhanced binding affinity for glycosaminoglycans (GAGs) [33,34] The G306E mutation in the E protein reduces the affinity between the virus and receptor, resulting in the loss of neurotoxicity of JEV [35]. The S156P substitution of the E completely abolished the transmissibility and restricted tissue tropism of TMUV in ducks [20]. In this study, introduction of K326E mutation significantly attenuated the neuroinvasiveness and neurovirulence of TMUV HB in mice. The 326K residue is located in domain III of the E protein, which is the receptor-binding region, suggesting that the K326E mutation might affect the receptor-binding pattern of the virus. Studies have shown that multiple amino acid mutations in the E protein influenced pathogenesis of TMUV in animal models [15,16,17]. The substitution of amino acids with different acidity and alkalinity in TMUV HB alter the binding affinity between the virus and glycosaminoglycans (GAGs), leading to changes in viral pathogenicity. In addition, the substitution of 326 amino acids may alter the hydrogen bond arrangement, leading to changes in the thermal stability and characteristics of the virus. This study will greatly enhance our understanding of the pathogenesis of TMUV infection in mammals and have implications for the development of a safe and efficient vaccine.

The evaluation of inflammatory cytokines in the mouse brain revealed that TMUV HB with E 326K induced higher levels of proinflammatory cytokines and type-I IFN, which could be a major reason for enhanced viral virulence and high mortality in mice. To further investigate the role of the type-I IFN in the virulence of TMUV, the infected mice were inoculated with 3000 U of IFN-β by i.c. at 3 dpi and 5 dpi. The results showed that the recombinant IFN-β accelerated the death of mice in the wtHB-E_326K_NS3_519T_ group and increased the mortality of mice in the smHB-E_326E_NS3_519T_ group compared to the untreated groups. All the data indicated that the overstimulated type-I IFN in the brain contributed to aggravating the disease in mice, apart from its antivirus function. Similarly, excessive IFN-α/β signaling in response to acute influenza virus infection led to uncontrolled inflammation and higher morbidity in mice [36]. Increasing evidence suggests that in viral infections, inappropriately regulated, excessive, or untimely type-I IFN responses can have deleterious effects [37]. Aberrantly regulated type-I IFN and excessive inflammatory responses have been shown to cause lethal pneumonia in SARS-CoV-infected mice [38]. Additionally, IFNs contribute to inflammatory responses after traumatic brain injury (TBI). Therefore, all the data suggest that type-I IFN is a double-edged sword in innate immunity, which can enhance the inflammation response and cause aggravated disease outcome. A delicate balance must therefore be struck where type-I IFN signaling must be sufficient to induce an adequate immune response, yet not so overwhelming as to induce immunopathology. Attempts should be made to develop small-molecule drugs that reduce interferon levels in the body or use interferon receptor blockers or modulators to maintain normal immune levels.

In immune cells, IRF7 is the primary regulator of type-I IFN signal transduction in immune cells and can induce the proinflammatory cytokine (IL-6) in pDCs and monocytes [38,39,40]. IRF7 is a key regulator of type-I IFN in combating pathogenic infections. Thus, tight regulation of IRF7 expression and activity is imperative in dictating appropriate type-I IFN production for normal IFN-mediated physiological functions. In our study, the wtHB-E_326K_NS3_519T_ infection significantly upregulated the expression of RIG-I compared with the smHB-E_326E_NS3_519T_ group, which further stimulated the transcription of a large amount of IRF7 and type-I IFN. This, in turn, led to the secretion of more type-I IFNs and the synthesis of more RIG-1 and IRF7 through positive feedback loops. However, excessive production of proinflammatory cytokines and interferon has been shown to exacerbate disease [38]. All data indicated that the 326K in the E protein plays an important role in inducting an inflammatory and IFN response in the brain.

Taken together, TMUV containing 326K provokes an excessive inflammatory and IFN response in the mouse brain through the RIG-I-IRF7 pathway, resulting in enhanced virulence. All results suggested that TMUVs might pose a potential threat to mammals.

## Figures and Tables

**Figure 1 viruses-15-02376-f001:**
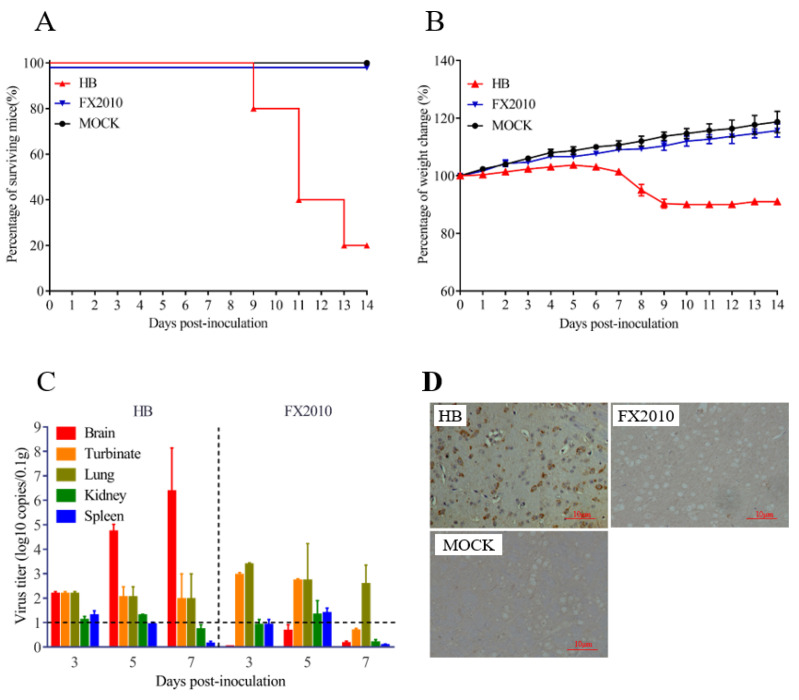
The TMUV HB strain exhibits high neuroinvasiveness in mice infected by i.n inoculation routes. (**A**) Neuroinvasiveness experiments in adult mice. Groups of five-week-old BALB/c mice (n = 5) were inoculated i.n. with 10^5.0^ TCID_50_ of HB and FX2010. Mock-infected mice received PBS. Mice were monitored daily for the appearance of symptoms during the 14-day period of observation. (**B**) Weight change of mice after challenge. After the challenge as described above, mouse body weights were recorded daily until the end of the experiment. Statistically significant differences in body weight were determined using two-way ANOVA multiple comparisons. A *p* value < 0.05 was considered significant. (**C**) Viral loads in different organs of mice determined by TaqMan probe-based qRT-PCR. (**D**) Immunohistochemical results of brain tissue in mice infected with TMUV.

**Figure 2 viruses-15-02376-f002:**
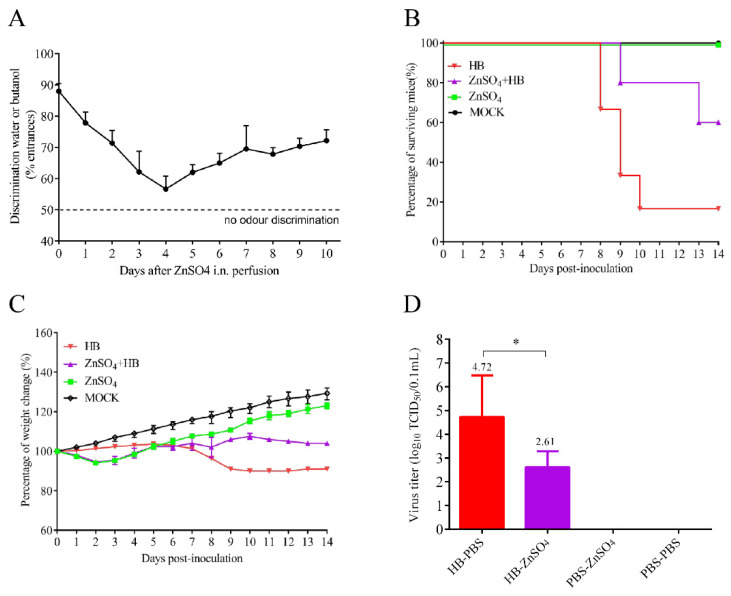
Olfactory neurons are the bridge for TMUV HB to invade the mouse brain. (**A**) Olfactory epithelium (OE) degeneration after ZnSO_4_ i.n. perfusion in five-week-old female BALB/c mice (n = 6). The discrimination ability between water and butanol was tested. The mean kinetic curves were determined by using (for each point) individual percentage scores for water and butanol obtained with the T-maze test procedure. (**B**) Partial destruction of olfactory epithelium reduced the mortality of mice. On day 4 post-perfusion, mice were inoculated with 30 μL containing 10^5.0^ TCID_50_ of HB via the i.n., while the HB group, the ZnSO_4_ group, and the MOCK group were set as control groups to record mouse survival. (**C**) Mice were monitored daily for symptoms and weight changes were recorded. (**D**) Viral load in the mouse brain. On day 7 post-inoculation, brains were collected and homogenized in PBS. Viral loads were measured with TCID_50_ assays in BHK-21 cells (*, *p* < 0.05).

**Figure 3 viruses-15-02376-f003:**
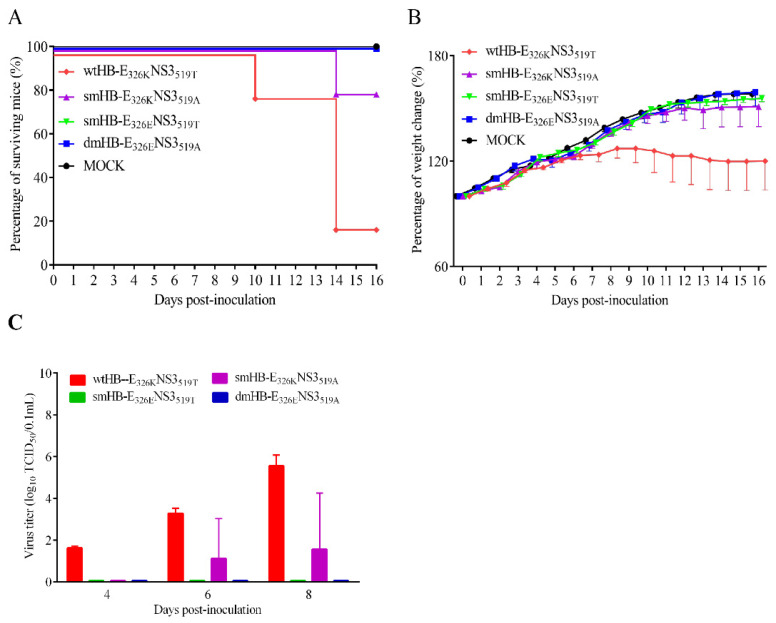
Survival rates and weight change curves of mice inoculated i.n. with wild type HB and mutant viruses. (**A**) Neuroinvasiveness of wild type HB and its mutants. Five-week-old female BALB/c mice were infected by the i.n. route with virus at a dose of 10^5.0^ TCID50 or MOCK infected and monitored for 16 days. (**B**) Weight changes of mice per day after challenge are shown. Statistically significant differences in body weight were determined using two-way ANOVA multiple comparisons. A *p* value < 0.05 was considered significant. (**C**) Three mice from each group were euthanized and the brains at 4, 6, and 8 dpi were collected and homogenized in PBS to yield 1:1 (mL/g) tissue homogenates, and clarified by centrifugation at 12,000× *g* rpm for 10 min at 4 °C. Viral loads were measured with TCID in BHK-21 cells.

**Figure 4 viruses-15-02376-f004:**
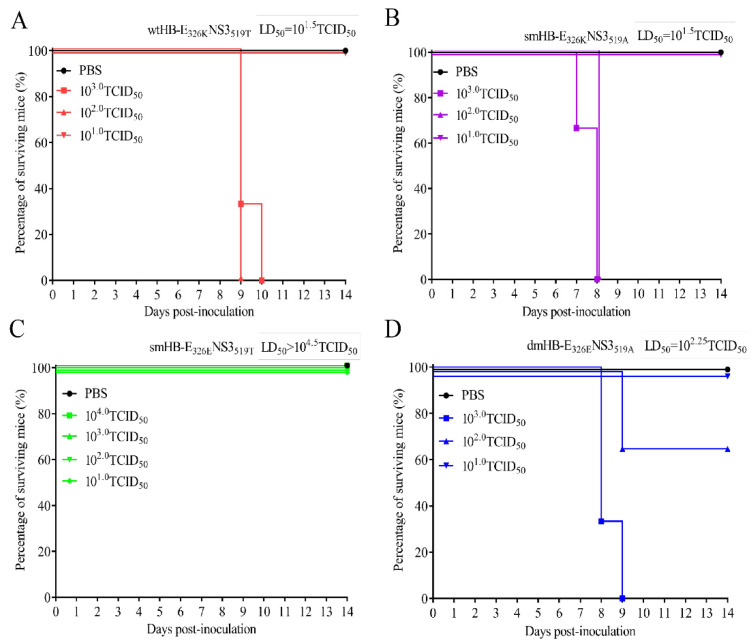
Identification of key amino acids affecting neurovirulence of wild type HB and mutant viruses. (**A**–**D**) Neurovirulence test in mice. Five-week-old BALB/c mice were infected by the i.c. route with four viruses at different doses (10^1.0^, 10^2.0^, 10^3.0^ or 10^4.0^ TCID_50_), respectively. Animals were monitored daily for the appearance of symptoms during the two-week period of observation. Median mouse lethality was calculated according to the Reed–Muench method.

**Figure 5 viruses-15-02376-f005:**
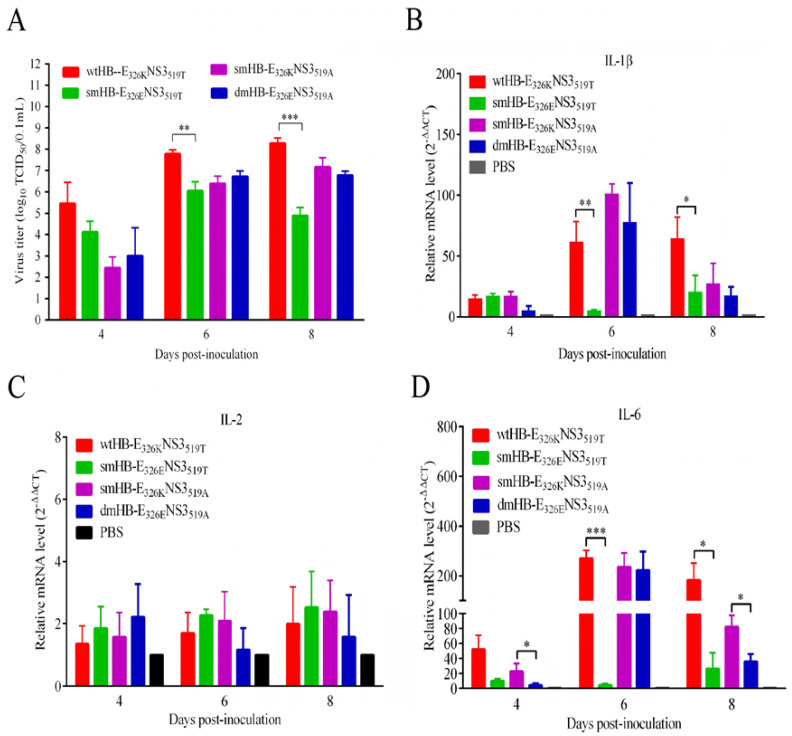
The K326E mutation reduces viral titers and attenuates inflammatory responses in the mouse brain. (**A**) Viral loads in the brain of mice. Five-week-old BALB/c mice were infected by the i.c. route with virus at a dose 10^3.0^ TCID_50_. Mouse brains were collected at 4, 6, and 8 dpi and homogenized in PBS to obtain supernatants. The replication titers in the supernatants were titrated with TCID_50_ assays in BHK-21 cells and tested by Student’s *t*-test. (**B**–**F**) Supernatants of infected mouse brain homogenates were collected for measurement of proinflammatory cytokines at mRNA levels by qRT-PCR. (**G**,**H**) IFN-α/β in the supernatants were determined by qRT-PCR, respectively. The significant differences at different time points are labeled (***, *p* < 0.001; **, *p* < 0.01; *, *p* < 0.05).

**Figure 6 viruses-15-02376-f006:**
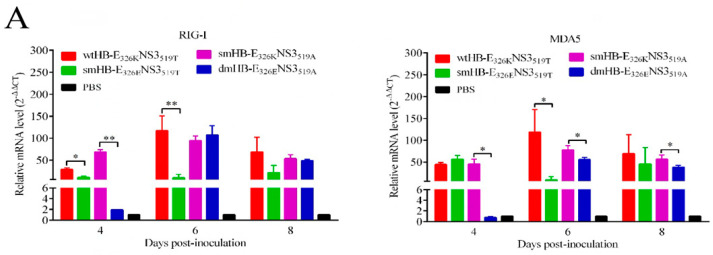
Amino acid 326K of the E protein plays a critical role in upregulating the RIG-I-IRF7 pathway. (**A**–**C**) The mRNA levels of RIG-I, MDA5, TLR3, TLR7, IRF3, and IRF7 in the brain of wtHB-E_326K_NS3_519T_-, smHB-E_326E_NS3_519T_-, smHB-E_326K_NS3_519A_-, and dmHB-E_326E_NS3_519A_-infected mice at the indicated time points were examined by using qRT-PCR (**, *p* < 0.01; *, *p* < 0.05).

## Data Availability

Data are contained within the article and Appendix A.

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
