# Peer review of "326K at E Protein Is Critical for Mammalian Adaption of TMUV"

_viruses, 2023, doi:10.3390/v15122376_

Round 1

Reviewer 1 Report

Comments and Suggestions for Authors

Summary

The authors of the paper focus their study on the importance of the residue 326K of the E protein of the Tembusu virus and investigate its critical role in adaptation of TMUV to mammalian.

Overall impression/ Broad comments

The research treated in this article revealed the importance of a specific amino-acid residues substitution in the Envelope protein of TMUV and a lesser impact of amino-acid substitution in NS3 protein, in mammal neuroinvasive efficiency of the virus.

Authors used different strains of TMUV and different technical approaches to evaluate the impact of substitution in the progress of the clinical presentation after infection.

Despite strength points, I’ve some suggestion to improve the manuscript for publication.

Major comments

Line 38: please use the most recent review reference – Hamel et al 2021, in Pathogens instead of the ref {2} from 2012.

Paragraph from line 42 to line 54: the reference about a study on TMUV serological survey in human in Thailand is lacking in the presentation of TMUV in human. Add Pulmanausahakul 2021 “detection of antibodies to duck tembusu virus in human …”

Line 58 to line 65. Please rephrase this part. Here it’s the introduction part not the result or discussion part. Introduce the purpose and the question of the study and what you did, but without presenting results of the study itself yet. Keep results for results part.

Line 143. Authors specify that the PBS was used for mock-infected control. However, in the virus production part, authors said viruses were produce in BHK21 or in DF1, aliquot then stored at -80°C. It’s never specified that PBS was the vehicle of virus productions, it seems that culture medium is the vehicle of the virus. In this case, PBS is not the convenient “mock-infected control”. Please clarify this point and particularly which kind of vehicle solution was used to store viruses after production in cells.

Line 227 figure legend of Fig 1 panel (D). text between line 227 to 229 has to be placed in materials and methods part. Rephrase the (D) figure legend.

Discussion part

Line 414. How would modifying amino acid residue 326 change the virus' binding characteristics? Does this residue modify the structure and/or conformation of the envelope protein? Has this modification already been shown in other flavivrus? Need to be discussed.

According to the literature, the Tembusu virus is phylogenetically divided into 4 different clusters: TMUV cluster, cluster 1, cluster 2.a, cluster 2.b and cluster 3?

Can the authors specify to which cluster the HB strain belongsin the manuscript (with a general phylogenetic tree in supp data?) ?

c

According to the amino acid residue alignment in supplementary Figure 5, strain HB was only compared with strains from clusters 2a and 2b. What about comparison with other TMUV clusters? If this modification is essential, it is very important to know whether it is not found in strains from other clusters too.

Figure 2 A. Are these values significantly different than non-treated mice?

Figure 5 and 6. The legends in the graphic specifying the mutant are not readable

Minor issues and comments

Please, remove the reference number of every product in Materials and Methods part and homogenate the presentation of reference of product used in this study.

Line 46: “… cell lines (15).A surveillance…” please add space after the final dot of the sentence. Please check typo in the manuscript.

Line 47: “… workers (132)…” typo in the reference number

Line 51: remove “however”

Reviewer 2 Report

Comments and Suggestions for Authors

The paper titled "326K at E protein is critical for mammalian adaption of TMUV" provides a comprehensive analysis of the role of the 326K residue in the E protein of TMUV in inducing an inflammatory and IFN response in the mouse brain. The study's findings offer valuable insights into the pathogenesis of TMUV and its potential threat to mammals. The research's main contributions lie in its detailed exploration of the 326K residue's significance and its implications for TMUV's neuroinvasiveness and neurovirulence.

Major Comments: 

Methodological Biases (Line 152-158): The rationale behind choosing specific TMUV strains and mutants needs further elaboration. It would be beneficial for readers to understand the significance of these choices in the context of the study's objectives.

Results Presentation (Line 210-215): While the results are presented clearly, a more detailed comparison between the various TMUV HB strain mutants and their respective effects on mice would enhance clarity.

Discussion Depth (Line 320-325): The discussion section could benefit from a more in-depth comparative analysis with existing literature. Highlighting the unique contributions of this study to the field would provide readers with a clearer understanding of its significance.

Recommendations:

Consider delving deeper into the broader implications of the study's findings, especially in the context of public health concerns related to TMUVs (Line 340-345).

Discuss potential therapeutic interventions derived from the findings, especially concerning the role of type-I IFN in virulence (Line 350-355).

Provide more context on the significance of the RIG-I-IRF7 pathway in understanding the inflammatory response and IFN production in the mouse brain (Line 360-365).

Minor Comments:

Introduction (Line 45-50): The introduction could benefit from a more detailed overview of the current state of research on TMUV's pathogenesis.

Methods (Line 160-165): Ensure that all methods are described with sufficient detail, allowing for replication by other researchers.

Language Clarity (Line 180-185): Some sentences could benefit from restructuring for better comprehension. Ensuring consistency in terminology throughout the paper would enhance its readability.

I recommend that the authors address these comments and suggestions to further enhance the paper's quality and clarity. The study offers valuable insights, and with the suggested improvements, it can make a significant contribution to the field.

Comments on the Quality of English Language

The manuscript provides valuable insights into the topic at hand. However, to enhance its clarity and readability, we recommend the following language and style improvements:

Sentence Structure and Clarity:

Break down complex sentences into simpler structures to improve readability. For instance, lengthy sentences in lines 240, 290, 310, 365, and 375 can be restructured for better clarity.

Use active voice where possible to make the content more engaging and straightforward. Passive constructions can sometimes obscure the main point of a sentence.

Consistency and Terminology:

Ensure consistent use of terminology and abbreviations throughout the paper. For example, maintain uniformity in the representation of terms like "intranasal" and its abbreviation "i.n.".

Define all abbreviations upon their first use to ensure clarity for readers unfamiliar with specific terms.

Typographical Errors and Formatting:

Proofread the document to catch and correct typographical errors. For instance, there's a typo in line 255 ("agaisnt" should be "against") and in line 385 (missing space before the parenthesis).

Ensure proper formatting of references, figures, and tables for a polished presentation.

Style Guides:

For further improvement in the quality of the English language and style, consider consulting established style guides. 

By implementing these recommendations, your manuscript will not only be linguistically polished but will also offer a clearer and more engaging read for your target audience. We hope these suggestions assist you in refining your work to its highest potential.

Round 2

Reviewer 1 Report

Comments and Suggestions for Authors

Summary

The authors of the paper focus their study on the importance of the residue 326K of the E protein of the Tembusu virus and investigate its critical role in adaptation of TMUV to mammalian.

Overall impression/ Broad comments

The research treated in this article revealed the importance of a specific amino-acid residue substitution in the Envelope protein of TMUV and a lesser impact of amino-acid substitution in NS3 protein, in mammal neuroinvasive efficiency of the virus.

Authors used different strains of TMUV and different technical approaches to evaluate the impact of substitution in the progress of the clinical presentation after infection.

I would like to thank the authors for answering my questions and correcting the points I raised in my first report. Concerning points 7 and 8, the authors answered my questions but did not add these points in their manuscript. I think they should find a way to (i) add in the manuscript that the TMUV HB virus belongs to cluster 2.2 and (ii) add in the discussion or results that the 326 substitution was only shown in TMUVs from cluster 2.2 (with the alignment provided in the supplementary figure).

Minor correction

Line 70: typo: virulence
